# New Methyl Threonolactones and Pyroglutamates of *Spilanthes acmella* (L.) L. and Their Bone Formation Activities

**DOI:** 10.3390/molecules25112500

**Published:** 2020-05-28

**Authors:** Retno Widyowati, Melanny Ika Sulistyowaty, Nguyen Hoang Uyen, Sachiko Sugimoto, Yoshi Yamano, Hideaki Otsuka, Katsuyoshi Matsunami

**Affiliations:** 1Department of Pharmacognosy and Phytochemistry, Faculty of Pharmacy, Airlangga University, Gedung Nanizar Zaman Joenoes, Kampus C Unair, Surabaya 60115, Indonesia; 2Graduate School of Biomedical & Health Sciences, Hiroshima University; 1-2-3 Kasumi, Minami-ku, Hiroshima 734-8551, Japan; melanny-i-s@ff.unair.ac.id (M.I.S.); d173259@hiroshima-u.ac.jp (N.H.U.); ssugimot@hiroshima-u.ac.jp (S.S.); yamano@hiroshima-u.ac.jp (Y.Y.); 3Department of Pharmaceutical Chemistry, Faculty of Pharmacy, Airlangga University, Gedung Nanizar Zaman Joenoes, Kampus C Unair, Surabaya 60115, Indonesia; 4Graduate School of Pharmacy, Yasuda Women’s University, Hiroshima 734-8551, Japan; otsuka-h@yasuda-u.ac.jp

**Keywords:** *Spilanthes acmella*, alkaline phosphatase, mineralization, methyl threonolactone, pyroglutamate

## Abstract

In our continuing research for bioactive constituents from natural resources, a new methyl threonolactone glucopyranoside (**1**), a new methyl threonolactone fructofuranoside (**2**), 2 new pyroglutamates (**3** and **4**), and 10 known compounds (**5**–**14**) were isolated from the whole plant of *Spilanthes acmella* (L.) L. The structures of these compounds were determined based on various spectroscopic and chemical analyses. All of the isolated compounds were evaluated on bone formation parameters, such as ALP (alkaline phosphatase) and mineralization stimulatory activities of MC3T3-E1 cell lines. The results showed that the new compound, 1,3-butanediol 3-pyroglutamate (**4**), 2-deoxy-d-ribono-1,4-lactone (**6**), methyl pyroglutamate (**7**), ampelopsisionoside (**10**), icariside B_1_ (**11**), and benzyl α-l-arabinopyranosyl-(1→6)-β-d-glucopyranoside (**12**) stimulated both ALP and mineralization activities.

## 1. Introduction

Osteoporosis is an age-related chronic disease characterized by a decrease of bone mineral density and an increased risk of bone fracture. As the population in the world ages, osteoporosis is becoming an important social problem. In the world, more than 200 million people are suffering from osteoporosis. Approximately 50% of women over 50 years old will have an osteoporosis-related fracture in their lifetime. An imbalance of bone remodeling causes osteoporosis through bone resorption by osteoclasts and bone formation by osteoblasts. Insufficient bone formation is an essential cause of osteoporosis. Mesenchymal stem cells differentiate to osteoblasts with activation of alkaline phosphatase (ALP) and bone mineralization, which are regulated by a variety of molecules such as Runt-related transcription factor 2 (Runx2), bone morphogenetic proteins (BMPs), and estrogen [1].

*Spilanthes* is a genus comprising over 60 species that are widely distributed in tropical and subtropical regions of the world, such as Africa, America, Borneo, India, Sri Lanka, and Asia [2,3]. *Spilanthes acmella* (L.) L. (Asteraceae) has commonly been used as a folk remedy, e.g., for toothaches, skin diseases, sexual deficiencies [4], rheumatism, fever, antioxidant [5], dysentery, snakebite, stammering in children [3], antiseptics, antibacterials, antifungals, antimalarials, influenza, cough, rabies, and tuberculosis [6,7]. It is also known to have been used as a panacea (Sumatra, Indonesia), the remedy of toothaches (Sudan), stomatitis (Java, Indonesia), and wound healing (India) [8].

The main constituents from the whole aerial parts, flower heads, and roots of this plant are spilanthol and acmellonate used to reduce toothaches, to induce saliva secretion [4,6,9], as powerful insecticides [10,11], and as local anesthetics [3]. It is also an important source of highly valuable bioactive compounds, such as phenolics, coumarins, triterpenoids [12], and flavonoids [13].

In our previous study, a combination of 70% ethanol extract of this plant and physical exercise increased testosterone level and osteoblast cell differentiation against glucocorticoid-induced osteoporotic male mice [14]. In addition, the 1-butanol and water layers of a 70% ethanol extract of this plant stimulated an osteoblast cell marker, ALP, of MC3T3-E1 osteoblast-like cells (126% and 127%, respectively) [15]. Based on these results, this plant extract seems to have the potential to be used as osteoporotic therapy by increasing bone formation. Therefore, it is important to know which compounds support these activities to understand the molecular basis and future development of the anti-osteoporotic remedy.

In this study, our further investigation on the chemical constituents of the 1-butanol layer demonstrated the presence of 2 new methyl threonolactone glycosides, 2-*C*-methyl-d-threono-1,4-lactone-3-*O*-β-d-glucopyranoside (**1**) and 2-*C*-methyl-d-threono-1,4-lactone-2-*O*-α-d-fructofuranoside (**2**); 2 new pyroglutamates, 1,3-butanediol 1-pyroglutamate (**3**) and 1,3-butanediol 3-pyroglutamate (**4**); and 10 known compounds, 2-*C*-methyl-d-threono-1,4-lactone (**5**) [16], 2-deoxy-d-ribono-1,4-lactone (**6**) [17], methyl pyroglutamate (**7**) [18], dendranthemoside A (**8**) [19], dendranthemoside B (**9**) [19], ampelopsisionoside (**10**) [20], icariside B_1_ (**11**) [21], benzyl α-l-arabinopyranosyl-(1→6)-β-d-glucopyranoside (**12**) [22], chicoriin (**13**) [23], and uridine (**14**) [24] (Figure 1). These compounds were isolated using various chromatographic techniques, such as silica gel, octadecylsilylated silica gel (ODS), and HPLC. The chemical structures were then determined by spectroscopic analyses using infrared (IR), high-resolution electrospray ionization mass spectrometry (HR-ESI-MS), 1D, and 2D NMR (Appendix A). Besides, the isolated compounds were evaluated on bone formation parameters. Among them, **4**, **6**, **7**, **10**, **11**, and **12** showed osteoblast stimulation activities.

## 2. Results and Discussion

### 2.1. Identification of Compounds

The 1-butanol layer of a methanol extract of *S. acmella* was fractionated with the guide of osteoblastic activity. As a result, 14 compounds (**1**–**14**) were isolated using several types of chromatography. Among the four new compounds, two methyl threonolactone were found to be glycosylated with a different monosaccharide, D-glucose, and D-fructose (**1** and **2**), respectively, and the other two were positional isomers of glutamate with 1,3-butanediol (**3** and **4**).

#### 2.1.1. Structure of 2-*C*-methyl-d-threono-1,4-lactone-3-O-β-d-glucopyranoside (**1**)

Compound **1** was obtained as a colorless amorphous powder with a molecular formula of C_11_H_18_O_9_ as determined by HR-ESI-MS at an *m/z* of 317.0845 [M + Na]^+^ (calcd for C_11_H_18_O_9_Na: 317.0843). The IR spectrum implied the presence of carbonyl (1777 cm^−1^), hydroxy groups (3392 cm^−1^), and C-O linkage (1210 and 1078 cm^−1^). The ^1^H-NMR spectrum (Table 1) displayed signals due to a methyl proton at *δ*_H_ 1.41 (3H, s); four oxygenated methylene protons at *δ*_H_ 3.60 (1H, dd, *J* = 11.6, 7.4 Hz), 3.91 (1H, dd, *J* = 11.6, 2.5 Hz), 4.10 (1H, dd, *J* = 9.5, 6.4 Hz), and 4.52 (1H, dd, *J* = 9.5, 6.5 Hz); five oxygenated methine protons at *δ*_H_ 3.238 (1H, dd, *J* = 9.2, 7.7 Hz), 3.237 (1H, dd, *J* = 9.8, 9.1 Hz), 3.34 (1H, ddd, *J* = 9.8, 7.4, 2.5 Hz), 3.36 (1H, t-like, *J* = 9.1 Hz), and 4.43 (1H, t-like, *J* = 6.5 Hz); and an anomeric proton at *δ*_H_ 4.37 (1H, d, *J* = 7.7 Hz) that indicated the presence of a glycosyl moiety. A lactone moiety was also indicated from three degrees of unsaturation, the relatively higher wavenumber of carbonyl (1777 cm^−1^), the esterified low field-shifted chemical shifts, and non-equivalency of H-5α (*δ*_H_ 4.10, 1H, dd, *J* = 9.5, 6.4 Hz) and H-5β (*δ*_H_ 4.52, 1H, dd, *J* = 9.5, 6.5 Hz).

The ^13^C-NMR spectrum (Table 2) of **1** showed 11 carbon resonances that were classified by chemical shift values and the HSQC spectrum into a methyl carbon (*δ*_C_ 19.2), two oxygenated methylene carbons (*δ*_C_ 63.2 and 69.6), five oxygenated methine carbons (*δ*_C_ 72.2, 74.9, 78.1, 78.2, and 83.4), a quaternary carbon (*δ*_C_ 75.7), an anomeric carbon (*δ*_C_ 104.0) and a carbonyl carbon at *δ*_C_ 179.4. The NMR spectroscopic data of **1** closely resembled that of 2-*C*-methyl-d-threono-1,4-lactone (**5**) [16,17], except for a large difference in the chemical shift values at C-4. The deshielded proton at *δ*_H_ 4.43 and the carbon at *δ*_C_ 83.4 of **1** suggested that the glycosyl group was attached to C-4. This was confirmed by an HMBC experiment from the correlation between H-4 (*δ*_H_ 4.43) and the carbon signal at *δ*_C_ 104.0 (Figure 2a). While five signals corresponding to C2′-C6′ and an anomeric carbon signal resonating at *δ*_C_ 104.0 were indicative of glucopyranoside. The relative configuration of the aglycone moiety was established using NOESY analysis. The correlations between H-5α/Me-6 and H-4/H-5β suggested α-orientation for H-5α and Me-6, and β-orientation for H-4 and H-5β (Figure 2b). Acid hydrolysis of **1** with 1N HCl liberated glucose and aglycone (**1a**). The glucose was determined to be D-series from the result of HPLC analysis with an optical rotation detector [25]. The coupling constant (*J* = 7.7 Hz) of the anomeric proton signal (*δ*_H_ 4.37) suggested a β-glycosidic linkage. The aglycone (**1a**) was identified to be 2-*C*-methyl-d-threono-1,4-lactone (**5**) [16] by spectroscopic (IR, HR-ESI-MS, 1D NMR, and [α]_D_) analyses. The 4*R* configuration was also supported by the application of the β-d-glucosylation-induced shift-trend rule, i.e., *Δδ*
_glucoside-aglycone_ values of C-3 (±0), C-4 (+7.4), and C5 (−3.4) suggested C-5 is Pro*-S* [26]. Based on these results, the structure of **1** was determined to be 2-*C*-methyl-d-threono-1,4-lactone-3-*O*-β-d-glucopyranoside.

#### 2.1.2. Structure of 2-*C*-methyl-d-threono-1,4-lactone-3-*O*-α-d-fructofuranoside (**2**)

Compound **2** was obtained as a colorless amorphous powder whose molecular formula was determined to be C_11_H_18_O_9_ from positive-ion HR-ESI-MS at an *m/z* of 317.0844 [M + Na]^+^ (calcd 317.0843). The ^1^H and ^13^C-NMR spectra (Table 1 and Table 2) of **2** were similar to those of **1** and 2-*C*-methyl-d-threono-1,4-lactone (**5**). The ^13^C-NMR also showed two secondary carbons at *δ*_C_ 60.6 and 62.9; three tertiary carbons at *δ*_C_ 79.0, 82.6, and 84.7; and a quaternary carbon at *δ*_C_ 109.2, indicating a fructofuranose moiety. Zhang et al. (2009) described that α and β orientations of D-fructofuranose were distinguishable by the *J* value of H-3 and the chemical shift of C-2. The characteristic values of α orientation are ^3^*J*_H3,H4_ = 3–4 Hz and C-2 (*δ*_C_ 107–109), while the β orientation is ^3^*J*_H3,H4_ = 7–9 Hz and C-2 (*δ*_C_ 103–106) [27], which suggested the α-fructofuranosylation to the aglycone. The 2D NMR analyses (COSY, HSQC, and HMBC) supported the planar structure of **2** (Figure 2a). The upfield-shift of C-4, compared to **1,** suggested the quarternary C-3/C-2′ connectivity. Acid hydrolysis of **2** yielded aglycone (2-*C*-methyl-d-threono-1,4-lactone) (**2a**) and D-fructose. Thus, **2** was estimated to be 2-*C*-methyl-d-threono-1,4-lactone-3-*O*- α -d-fructofuranoside, although a chemical synthesis may support our conclusion.

#### 2.1.3. Structure of 1,3-butanediol-1-pyroglutamate (**3**)

Compound **3** was obtained as a colorless amorphous powder and displayed an [M + Na]^+^ ion at an *m/z* of 224.0890 (calcd 224.0893) corresponding to a molecular formula of C_9_H_15_O_4_N. The IR spectrum showed a strong absorption band for the hydroxy (3331 cm^−1^) and carbonyl (1735 and 1684 cm^−1^). The ^1^H and ^13^C-NMR spectra of **3** showed signals assignable to two methylenes (*δ*_H_ 2.31 (1H, ddd, *J* = 17.1, 9.9, 5.5 Hz)) and *δ*_H_ 2.37 (1H, ddd, *J* = 17.1, 9.4, 7.3 Hz) / *δ*_C_ 30.4 (C-3), and *δ*_H_ 2.16 (1H, m) and 2.48 (1H, m)/*δ*_C_ 26.0 (C-4)), a methine (*δ*_H_ 4.29 (1H, dd, *J* = 9.1, 4.4 Hz)/*δ*_C_ 57.3 (C-5)), and two carbonyls (*δ*_C_ 181.2 (C-2) and 174.2 (C-6)) (Table 1 and Table 2). The chemical shift values and coupling patterns of these signals suggested a pyroglutamate moiety [18]. In addition, the ^1^H and ^13^C-NMR spectra also revealed a 1,3-butanediol framework, with a methyl (*δ*_H_ 1.20 (3H, d, *J* = 6.3 Hz)/*δ*_C_ 23.9 (Me-10)), a methylene (*δ*_H_ 1.75 (1H, m) and 1.79 (1H, m)/*δ*_C_ 38.8 (C-2′)), an oxygenated methine [*δ*_H_ 3.86 (1H, dqd, *J* = 8.2, 6.3, 4.4 Hz)/*δ*_C_ 65.5 (C-3′)), and an oxygenated methylene (*δ*_H_ 4.27 (2H, m)/*δ*_C_ 64.0 (C-1′)). The relatively lower chemical shift value of H-1′ (*δ*_H_ 4.27) indicated the ester linkage with pyroglutamate at C-1′. The ^1^H–^1^H COSY spectrum displayed correlations through H-3, H-4, and H-5, and through H-1′, H-2′, H-3′, and H-4′. The HMBC spectrum demonstrated correlations of C-2 with H-3, and H-4, and correlations of C-6 with H-4 and H-5. Furthermore, the strong correlation of H-1′ with C-6 established that the 1,3-butanediol moiety was located at the C-6 (Figure 3) to form an ester linkage with primary alcohol. Therefore, the structure of **3** was elucidated to be 1,3-butanediol-1-pyroglutamate.

#### 2.1.4. Structure of 1,3-butanediol 3-pyroglutamate (4)

Compound **4** was isolated as a colorless amorphous powder. The NMR spectra together with the molecular ion at an *m/z* 224.0893 [M + Na]^+^ (calcd for C_9_H_15_O_4_NNa: 224.0893) in HR-ESI-MS indicated that **4** was an isomer of **3**. The shielding of H-1′ (*δ*_H_ 3.65 (1H, dt-like, *J* = 10.6, 6.3 Hz) and *δ*_H_ 3.68 (1H, dt-like, *J* = 10.6, 6.7 Hz)/C-1′ (*δ*_C_ 60.5)) and deshielding of H-3′ (*δ*_H_ 3.89 (1H, dqd, *J* = 7.7, 6.3, 4.8 Hz)/C-3′ (*δ*_C_ 65.5)) suggested that the ester linkage of 1,3-butanediol must be changed from primary alcohol (C-1′) to the secondary alcohol (C-3′) of 1,3-butanediol in **4** (Table 1 and Table 2). The planar structure was supported by COSY and HMBC correlations (Figure 3). The difficulty of observing the correlation between C-3′ and C-6 coincided with the abundant multiplicity of H-3′. As a result, the structure of **4** was estimated to be 1,3-butanediol 3-pyroglutamate.

The absolute stereochemistry of compounds **3** and **4** remains to be elucidated because of the insufficient amount for further analysis.

### 2.2. Osteoblast Activity

Osteoblasts are the most important cells in bone tissue and are critical for bone formation through proliferation and differentiation. During osteoblast differentiation, BMPs induce the expression of osteoblastic markers, such as ALP. Proliferating osteoblasts show ALP activity, which is greatly enhanced during in vitro bone formation. ALP is a membrane-bound enzyme that is often used as a marker for osteogenic differentiation. 17β-estradiol has a significant impact on bone mineral metabolisms. It affects osteoblast proliferation through modulating the release of several local regulators of bone turnover from monocytes and enhanced BMP-4 induced osteoblastic marker expression and mineralization [28].

To evaluate the effects of **1**–**14** on osteoblast function, ALP activity, which is related to the osteoid formation and initiation of the deposition of minerals, was evaluated. In this study, it was found that **4**, **6**, **7**, **10**, **11**, and **12** stimulated ALP activity, which markedly increased osteoblast growth and differentiation of osteoblastic MC3T3-E1 cells. Compounds **7** and **11** did not show concentration dependency, probably due to the toxicity at higher concentrations. At concentrations of 25 µM, **6**, **10**, and **12** stimulated ALP activity up to 112% comparable to the positive control, 17β-estradiol at 0.02 and 0.01 µM (Figure 4).

Osteoblasts can be induced to produce vast extracellular calcium deposition in vitro. This process is called mineralization. Calcium deposition is an indication of successful in vitro bone formation and can specifically be stained bright orange-red using Alizarin Red S. The effects of **1**–**14** were then examined by measuring the calcium deposition by Alizarin Red staining. As was found for the ALP activity study above, **4**, **6**, **7**, **10**, **11**, and **12** showed stimulatory effects on mineralization. Compounds **6**, **10**, and **12** stimulated the mineralization up to 112% at 25 μM, comparable to that of the positive control, 17β-estradiol, at 0.02 and 0.01 μM (Figure 5).

In bone formation, osteoblasts are key cells in bone matrix formation and calcification. Osteogenesis starts with osteoblast production and secretion of type I collagen, which makes up about 90% of the organic bone matrix or the osteoid. Osteoblast also becomes high in alkaline phosphatase. Alkaline phosphatase is released into the osteoid to initiate the deposition of minerals. After mineralization, the bone becomes hard and rigid with necessary mechanical properties to withstand the external forces to support the body and protect the internal organs. Our study demonstrated that **4**, **6**, **7**, **10**, **11**, and **12** stimulated both ALP activity and calcium deposition in osteoblastic MC3T3-E1 cell in vitro, which suggests that the extract of *S. acmella* and these compounds have potential to be a remedy for osteoporosis as osteoblastic bone formation stimulant.

## 3. Materials and Methods

### 3.1. General Experimental Procedures

^1^H and ^13^C-NMR spectra were taken on a Bruker Ultrashield Avance 600 spectrometer at 600 MHz and 150 MHz, respectively, with TMS as an internal standard. IR and UV spectra were measured on a HORIBA FT-720 FT-IR spectrophotometer and JASCO V-520 UV–vis spectrophotometer, respectively. Optical rotation was measured on a JASCO P-1030 digital polarimeter. Positive ion HR-ESI-MS was recorded using an LTQ Orbitrap XL mass spectrometer (Thermo Fisher Scientific, Waltham, MA, USA). Silica gel open column chromatography (CC) and reversed-phase (ODS) CC were performed on silica gel 60 (E. Merck, Darmstadt, Germany), and Cosmosil 75C18-OPN (Nacalai Tesque, Kyoto, Japan; Φ = 35 mm, L = 350 mm), respectively. HPLC was performed on an ODS column (Inertsil ODS-3, GL Science, Tokyo, Japan; Φ = 6 mm, L = 250 mm, 1.5 mL/min), and the eluate was monitored with a JASCO RI-930 intelligent detector and a JASCO PU-1580 intelligent pump unless otherwise specified.

### 3.2. Plant Material

Whole plants of *Spilanthes acmella* (L.) L. were collected in late June 2007 in Kebun Raya Purwodadi, Malang, Indonesia (07°46′09″–07°47′23″ South latitude and 112°16′23″–112°17′17″ East Longitude), and voucher specimens were deposited at the Department of Pharmacognosy and Phytochemistry, Faculty of Pharmacy, Airlangga University as SA30062007 [29,30].

### 3.3. Extraction and Isolation

The air-dried plants (2.0 kg) were extracted with methanol (MeOH, 10.0 l × 3). The methanol solution was concentrated and adjusted to 95% aq. methanol by the addition of water and then partitioned with *n*-hexane (1.0 l × 3, 23.5 g). The remaining aqueous methanol layer was evaporated and resuspended in 0.5 l of water and then partitioned with ethyl acetate (1.0 l × 3, 40.4 g) and 1-butanol (1.0 l × 3, 47.5 g), successively.

The 1-butanol layer (40.0 g) was subjected on silica gel (300 g) CC with increasing amounts of MeOH in CHCl_3_ [Hexane-CHCl_3_ (1:1), 4 l, CHCl_3_-MeOH (50:1, 40:1, 30:1, 20:1, 15:1, 10:1, 7:1, 5:1, 3:1, 2:1, 2 l), 500 mL fractions being collected], yielding 12 fractions (Fr. Sab1–Sab12). The fraction Sab11 (2.75 g) was subjected to ODS CC in 10% aq. MeOH (400 mL)–100% MeOH (400 mL), linear-gradient, led 10 fractions (Fr. Sab11-1–Sab11-10). The fraction Sab11-3 (442 mg) and Sab11-4 (123 mg) were purified by HPLC with 35% aq. MeOH to give **1** (10.1 mg) and **8** (dendranthemoside A, 3.71 mg), respectively. The fraction Sab10 (1.81 g) was subjected to ODS CC in 10% aq. MeOH (400 mL)–100% MeOH (400 mL) and led seven fractions (Fr. Sab10-1–Sab10-7). The fraction Sab10-1 (770 mg) was purified by HPLC with 100% H_2_O, YMC Triart C18 column. Three peaks appeared at 5, 18, and 35 minutes and were collected to give **2** (7.62 mg), **5** (2-*C*-methyl-d-threono-1,4-lactone, 9.31 mg), and **14** (uridine, 27.5 mg). The fraction Sab10-2 (193 mg) was purified by HPLC (20% aq. MeOH) to give **11** (icariside B_1_, 6.12 mg) and **13** (chicoriin, 2.99 mg). The fraction Sab10-3 (142 mg) was purified by HPLC (35% aq. MeOH) to give **9** (dendranthemoside B, 4.31 mg). The fraction Sab5 (710 mg) was subjected to ODS CC in 10% aq. MeOH (400 mL)–100% MeOH (400 mL) and led 10 fractions (Fr. Sab5-1–Sab5-10). The fraction Sab5-1 (483 mg) and Sab5-2 (68.3 mg) were purified by HPLC (100% H_2_O, YMC Triart C18 column) to give **3** (7.80 mg), **4** (4.21 mg), and **7** (methyl pyroglutamate, 6.63 mg), respectively. The mixture of fraction Sab6, Sab7, Sab8, and Sab9 (2.06 g) was subjected to ODS CC in 10% aq. MeOH (400 mL)–100% MeOH (400 mL) and led 10 fractions (Fr. Sab6-9-1–Sab6-9-10). The fraction Sab6-9-1 (340 mg) was purified by HPLC (100% H_2_O, YMC Triart C18 column) to give **6** (2-deoxy-d-ribono-1,4-lactone, 6.01 mg). The fraction, Sab6-9-4 (114 mg), was purified by HPLC (40% aq. MeOH) to give **10** (ampelopsisionoside, 5.43 mg). The fraction Sab12 (5.36 g) was subjected to ODS CC in 10% aq. MeOH (400 mL)–100% MeOH (400 mL) and led 10 fractions (Fr. Sab12-1–Sab12-10). The fraction Sab12-3 (129 mg) was purified by HPLC (35% aq. MeOH) to give **12** (benzyl α-l-arabinopyranosyl (1→6)-β-d-glucopyranoside, 4.51 mg).

#### 3.3.1. 2-*C*-methyl-d-threono-1,4-lactone-4-*O*-β-d-glucopyranoside (**1**)

Colorless amorphous powder; [α]_D_
^26.7^ -28.6 (*c* 0.78, MeOH); IR (film) *ⱱ*_max_ cm^−1^: 3392, 2924, 1777, 1713, 1650, 1557, 1456, 1391, 1210, 1078, 899, 647; ^1^H-NMR and ^13^C-NMR, see Table 1 and Table 2; positive HR-ESI-MS (*m/z*): 317.0845 [M + Na]^+^(calcd for C_11_H_18_O_9_Na: 317.0843).

#### 3.3.2. 2-*C*-methyl-d-threono-1,4-lactone-3-*O*-α-d-fructofuranoside (**2**)

Colorless amorphous powder; [α]_D_
^2^^6.9^ -10.4 (*c* 0.74, MeOH); IR (film) *ⱱ*_max_ cm^−1^: 3386, 2938, 1774, 1732, 1651, 1540, 1456, 1339, 1206, 1073, 870, 669; ^1^H-NMR and ^13^C-NMR, see Table 1 and Table 2; positive HR-ESI-MS (*m/z*): 317.0844 [M + Na]^+^(calcd for C_11_H_18_O_9_Na: 317.0843).

#### 3.3.3. 1,3-butanediol-1-pyroglutamate (**3**)

Colorless amorphous powder; [α]_D_
^27.7^ −0.86 (*c* 0.42, MeOH); IR (film) *ⱱ*_max_ cm^−1^: 3331, 2966, 2926, 1735, 1684, 1557, 1457, 1338, 1207, 1052, 670; ^1^H-NMR and ^13^C-NMR, see Table 1 and Table 2; positive HR-ESI-MS (*m/z*): 224.0890 [M + Na]^+^(calcd for C_9_H_15_O_4_NNa: 224.0893).

#### 3.3.4. 1,3-butanediol-3-pyroglutamate (**4**)

Colorless amorphous powder; [α]_D_
^27.1^ +1.40 (*c* 0.41, MeOH); IR (film) *ⱱ*_max_ cm^−1^: 3314, 2969, 2931, 1735, 1683, 1557, 1457, 1338, 1229, 1054, 669; ^1^H-NMR and ^13^C-NMR, see Table 1 and Table 2; positive HR-ESI-MS (*m/z*): 224.0893 [M + Na]^+^ (calcd for C_9_H_15_O_4_NNa: 224.0893).

#### 3.3.5. Acid Hydrolysis for Identification of Sugar Moiety of **1** and **2**

A solution of **1** or **2** (1 mg) in 1 N aq. HCl (0.1 mL) was heated at 80 °C for 2 h. The mixture was neutralized by the addition of amberlite IRA400 (OH^-^ form), and the resin was removed by filtration. Then, the filtrates were extracted with EtOAc. The aqueous layers were subjected to HPLC analysis (column: Shodex Asahipak NH 2P-50 4E, 250 × 4.6 mm i.d.; mobile phase: 75% CH_3_CN in water; detection: optical rotation (JASCO OR-2090Plus); flow rate: 1.0 mL/min) to identify D-glucose and D-fructose, which were identified by the comparison of their retention times with those of authentic samples; *t*_R_: 5.11 (D-fructose, negative optical rotation) and *t*_R:_ 6.10 (D-glucose, positive optical rotation). They also yielded the aglycon of 2-*C*-methyl-d-threono-1,4-lactone (**1a** and **2a**) [26]. (**1a**): Colorless amorphous powder; [α]_D_
^27.5^ −15.0 (*c* 0.06, MeOH); IR*ⱱ*_max_ (film) cm^−1^: 3286, 2923, 1729, 1649, 1558, 1456, 1339, 1205, 1092, 667; ^1^H-NMR (methanol-*d_4_*, *δ*_H_): 1.35 (3H, s, H-6), 3.96 (1H, dd, *J* = 9.0, 4.0 Hz, H-5a), 4.17 (1H, dd, *J* = 5.0, 4.0 Hz, H-4), 4.49 (1H, dd, *J* = 9.0, 5.0 Hz, H-5b); positive HR-ESI-MS *m/z* 155.0312 [M + Na]^+^(calcd. for C_5_H_8_O_4_Na: 155.0315) and *t*_R_ = 10.20 (HPLC, 100% H_2_O, YMC Triart C18 column). (**2a**): Colorless amorphous powder; [α]_D_
^27.9^ −15.7 (*c* 0.08, MeOH); IR*ⱱ*_max_ (film) cm^−1^: 3373, 2927, 1717, 1649, 1558, 1455, 1338, 1207, 1098, 668; ^1^H-NMR (methanol-*d_4_*, *δ*_H_): 1.35 (3H, s, H-6), 3.96 (1H, dd, *J* = 9.0, 4.0 Hz, H-5a), 4.17 (1H, dd, *J* = 5.0, 4.0 Hz, H-4), 4.49 (1H, dd, *J* = 9.0, 5.0 Hz, H-5b); positive HR-ESI-MS *m/z* 155.0319 [M + Na]^+^ (calcd. for C_5_H_8_O_4_Na: 155.0315) and *t*_R_ = 10.20 (HPLC, 100% H_2_O, YMC Triart C18 column) together with authentic sample **5**.

### 3.4. Osteoblast Activities

#### 3.4.1. Cell Culture

An osteoblast-like cell of MC3T3-E1 was purchased from Riken Cell Bank, Tsukuba, Japan. The cells were cultured in α-MEM medium containing 10% FBS in a CO_2_ incubator at 37 °C and sub-cultured every 3 days by trypsin (0.25%) treatment. The 5 × 10^4^ cells were seeded in 24-well plates and incubated for the following ALP and mineralization assays.

#### 3.4.2. Alkaline Phosphatase (ALP) Activity

The cells were treated at 90% confluence with the culture medium containing 10 mM β-glycerophosphate and 50 µg/mL ascorbic acid to initiate in vitro proliferation. The medium was changed every 2–3 d. After 6 days, the cells were cultured individually for 3 days with medium containing 0.3% bovine serum and isolated compounds (**1**–**14**). On harvesting, the medium was removed, and the cell monolayer was gently washed twice with phosphate-buffered saline. The cells were lysed with 0.2% Triton X-100, and centrifuged at 14,000 × g for 5 min. The clear supernatant was used to measure ALP activity using p-nitrophenylphosphate [31,32].

#### 3.4.3. Mineralization of MC3T3-E1

The cells were treated, at 90% confluence, with culture medium containing 10 mM β-glycerophosphate and 50 µg/mL ascorbic acid, to initiate in vitro mineralization. After 12 days, the cells were cultured individually for 2 days with medium containing 0.3% bovine serum and isolated compounds (**1**–**14**). On harvesting, the cells were fixed with 70% ethanol for 1 hour and then stained with 40 mM Alizarin Red S for 10 min with gentle shaking. To quantify the bound dye, the stain was solubilized with 10% cetylpyridinium chloride by shaking for 15 min. The absorbance of the solubilized stain was measured at 561 nm [33].

## 4. Conclusions

The chemical investigation of the 1-butanol layer of the methanol extract of *Spilanthes acmella* (L.) L. gave 14 compounds (**1**–**14**), including 2 new methyl threonolactone glycosides, 2-*C*-methyl-d-threono-1,4-lactone-3-*O*-β-d-glucopyranoside (**1**) and 2-*C*-methyl-d-threono-1,4-lactone-2-*O*-α-d-fructofuranoside (**2**); 2 new pyroglutamates, 1,3-butanediol 1-pyroglutamate (**3**) and 1,3-butanediol 3-pyroglutamate (**4**); and 10 known compounds, 2-*C*-methyl-d-threono-1,4-lactone (**5**), 2-deoxy-d-ribono-1,4-lactone (**6**), methyl pyroglutamate (**7**), dendranthemoside A (**8**), dendranthemoside B (**9**), ampelopsisionoside (**10**), icariside B_1_ (**11**), benzyl α-l-arabinopyranosyl-(1→6)-β-d-glucopyranoside (**12**), chicoriin (**13**), and uridine (**14**). All the isolated compounds were evaluated for ALP and calcium deposition as markers for osteogenic differentiation and mineralization stimulatory activities. Our study demonstrated that compounds **4**, **6**, **7**, **10**, **11**, and **12** stimulated ALP activity and calcium deposition in osteoblastic MC3T3-E1 cell in vitro. Compounds **6**, **10**, and **12** stimulated ALP and mineralization up to 112% at 25 µM, comparable to that of the positive control, 17β-estradiol, at 0.02 and 0.01 μM. Of these active compounds, compounds **4**, **6**, **7**, **10**, and **11** are the first time to be reported their osteoblastic activity. While compound **12** has already isolated as an osteoblastic active compound from the fruit of *Prunus mume* [34]. The isolation of the same active compound (**12**) from the different plants through independent study strengthened and supported the reliability of our results. Overall, the compounds **4**, **6**, **7**, **10**, **11**, and **12** are the active principle of *S. acmella* on the stimulation of osteoblastic bone formation and may play an important role in bone remodeling as drug seeds in osteoporosis therapy.

## Figures and Tables

**Figure 1 molecules-25-02500-f001:**
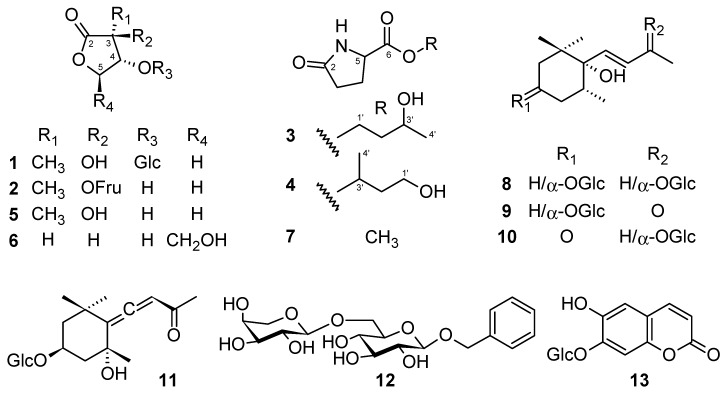
Structures of compounds **1**–**1****3**.

**Figure 2 molecules-25-02500-f002:**
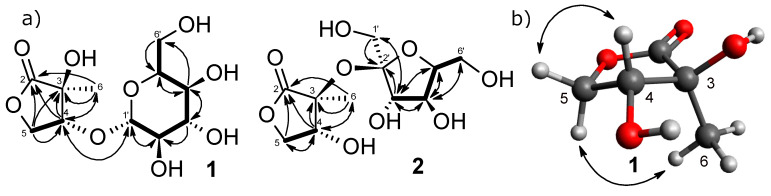
(**a**) 
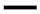

^1^H–^1^H COSY and 
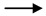
 HMBC correlations; (**b**) NOESY correlations.

**Figure 3 molecules-25-02500-f003:**
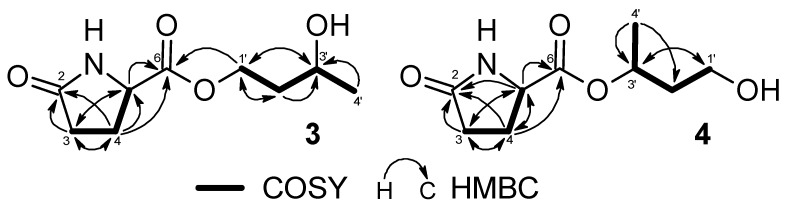
HMBC and COSY correlations of **3** and **4**.

**Figure 4 molecules-25-02500-f004:**
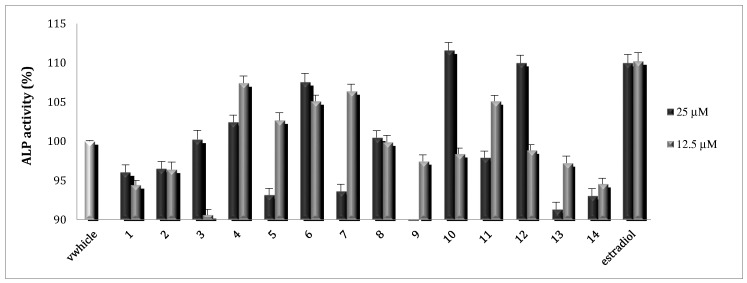
ALP activity of **1**–**14** toward MC3T3-E1 cell lines.

**Figure 5 molecules-25-02500-f005:**
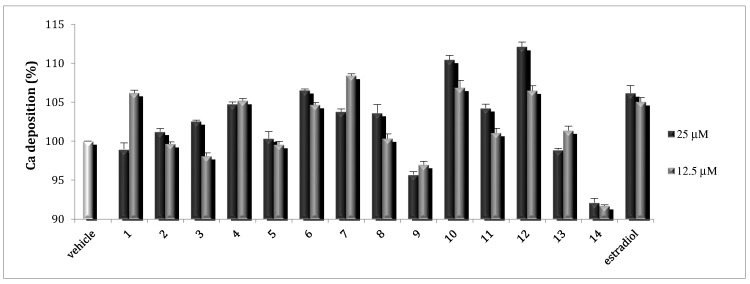
Calcium deposition of **1–14** toward MC3T3-E1 cell lines.

**Table 1 molecules-25-02500-t001:** ^1^H-NMR spectroscopic data for compounds **1**–**4**.

Position	1	2	3	4
3	-	-	2.31 ddd (17.1, 9.9, 5.5) 2.37 ddd (17.1, 9.4, 7.3)	2.32 ddd (16.9, 9.7, 5.6) 2.37 ddd (16.9, 9.4, 7.1)
4	4.43 t-like (6.5)	4.17 dd (5.5, 4.4)	2.16 m 2.48 m	2.17 dddd (13.0, 9.4, 5.6, 4.7) 2.49 dddd (13.0, 9.7, 9.2, 7.1)
5	4.10 dd (9.5, 6.4, α) 4.52 dd (9.5, 6.5, β)	3.97 dd (9.4, 4.4, α) 4.48 dd (9.4, 5.5, β)	4.29 dd (9.1, 4.4)	4.24 dd (9.2, 4.7)
6	1.41 s (3H)	1.35 s (3H)	-	-
1′	4.37 d (7.7)	3.63 d (12.1) 3.71 d (12.1)	4.27 m (2H)	3.65 dt-like (10.6, 6.3) 3.68 dt-like (10.6, 6.7)
2′	3.238 dd (9.2, 7.7)	-	1.75 m 1.79 m	1.62 dtd (13.8, 6.7, 4.8) 1.67 ddt (13.8, 7.7, 6.3)
3′	3.36 t-like (9.1)	4.04 d (4.2)	3.86 dqd (8.2, 6.3, 4.4)	3.89 dqd (7.7, 6.3, 4.8)
4′	3.237 dd (9.8, 9.1)	3.89 dd (6.4, 4.2)	1.20 d (3H, 6.3)	1.18 d (3H,6.3)
5′	3.34 ddd (9.8, 7.4, 2.5)	3.84 ddd (6.4, 4.9, 3.1)		
6′	3.60 dd (11.6, 7.4) 3.91 dd (11.6, 2.5)	3.64 dd (11.9, 4.9) 3.78 dd (11.9, 3.1)		

600 MHz (methanol-*d_4_*). Chemical shifts (*δ*) in ppm. m: multiplet or overlapped signals.

**Table 2 molecules-25-02500-t002:** ^13^C-NMR spectroscopic data for compounds **1**–**4**.

Position	1	2	3	4	1a
2	179.4	179.3	181.2	181.3	180.2
3	75.7 (±0) *	75.9	30.4	30.6	75.7
4	83.4 (+7.4) *	75.6	26.0	26.3	76.0
5	69.6 (−3.4) *	72.9	57.3	57.4	73.0
6	19.2	17.6	174.2	176.2	17.7
1’	104.0	60.6	64.0	60.5	
2’	74.9	109.2	38.8	42.6	
3’	78.2	82.6	65.5	66.3	
4’	72.2	79.0	23.9	23.9	
5’	78.1	84.7			
6’	63.2	62.9			

150 MHz (methanol-*d_4_*). Chemical shifts (*δ*) in ppm. *: Δδ _glucoside - aglycone._

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
