# Peer review of "New Methyl Threonolactones and Pyroglutamates of Spilanthes acmella (L.) L. and Their Bone Formation Activities"

_molecules, 2020, doi:10.3390/molecules25112500_

Round 1

Reviewer 1 Report

Four new compounds and ten known compounds were isolated from the 1-butanol layer of methanol extract of the air-dried plants of Spilanthes acmella. the structures of new compounds were well elucidated, alkaline phosphatase and mineralization stimulatory activities of MC3T3-E1 cell lines of all isolates were assayed.
Just several minor concerns:
1. It's better to supply the voucher Number of Spilanthes acmella.
2. "Materials and Methods" part,  authors declared "1H and 13C NMR spectra were taken on a Bruker Ultrashield Avans 600 spectrometer at 600 MHz and 150 MHz". However, the 13C NMR was declared as 125 MHz in Table 2.
3. Figure 2. It's better to supply legends for 1H-1H COSY and HMBC correlations.

Author Response

Dear Editor and Reviewers,

 Thank you for your useful and kind advice. We have investigated our manuscript again, according to the reviewer’s comments. The correction was indicated in red with yellow highlight.

 Reviewer 1:

Just several minor concerns:

  1. It's better to supply the voucher Number of Spilanthes acmella.

Thank you for the advice, the voucher number was added as SA30062007

  1. "Materials and Methods" part, authors declared "1H and 13C NMR spectra were taken on a Bruker Ultrashield Avans 600 spectrometer at 600 MHz and 150 MHz". However, the 13C NMR was declared as 125 MHz in Table 2.

We apologize for our mistake. The correct frequency was 150 MHz

  1. Figure 2. It's better to supply legends for 1H-1H COSY and HMBC correlations.

Thank you for the advice. We revised as follows to provide bold line and arrow to clarify the information on this figure;

Figure 2. a)          1H-1H COSY and         HMBC correlations  b) NOESY correlations.

Reviewer 2:

  1. Introduction section - the Authors should try to make an effort to emphasize the importance of their studies. They can also dedicate part of Introduction about the activity investigated in the study.

Thank you for the useful advice. We have added the importance of this study and about activity in the introduction section.

  1. The correct name of studied species is Spilanthes acmella (L.) L.

Thank you for your correction. We have revised it as the comment.

  1. Results and Discussion is very important part of each manuscript published. In presented manuscript this section is poor and comprises too general explanations. Authors should discuss their results with other scientific papers.

Thank you for your useful comments. We revised it to add importance and discussion with other scientific papers.

  1. Plant material - Please add the geographical coordinates of occurrence of the tested species. Which reference flora was used to identify the species?

Thank you for the useful comment. We added the geographical coordinates,  07°46’09”-07°47’23” South latitude and 112°16’23”-112°17’17” East Longitude, and the references to identify the plant.

  1. The HPLC conditions are needed.

We added additional information for the HPLC in General Experimental Procedures. We used this column condition mainly and added additional information in the separation section if we used another column.

  1. The conclusions should be integrated with more detailed results summarizing all the study and must reflect the innovation of this study and the perspectives.

 Thank you for your kind suggestions. We revised it to include innovation and perspectives.

Reviewer 3.

  1. Pages 2~8:

Numbers of the compounds → bold

2-C-methyl-d-threono-1,4-lactone-3-O-β-d- glucopyranoside (1) → 2-C-methyl-d-threono-1,4-lactone-3-O-β-d- glucopyranoside (1)

2-C-methyl- d-threono-1,4-lactone-3-O- β-d- fructofuranoside (2) →2-C-methyl- d-threono-1,4-lactone-3-O-β-d- fructofuranoside (2)

Thank you for the kind advice. We have corrected them.

  1. Page 3: Table 1

compound 1: 1.41 (3H, s) → 1.41 s (3H)

compound 2: 1.35 (3H, s) → 1.35 s (3H)

compound 3: 2.16 (m), 2.48 (m), 4.27 (2H, m), 1.75 (m), 1.79 (m) → 16 m, 2.48 m, 4.27 m (2H), 1.75 m, 1.79 m

Thank you for your correction. We are sorry to make careless mistake. We revised them.

  1. Page 3, line 103: The original reference should be cited instead of ref [25].

Thank you for the advice. We have added the original reference and changed the order of the reference number.

Kasai, R.; Suzuno M.; Asakawa J.; Tanaka O. Carbon-13 chemical shifts of isoprenoid-β-D-glucopyranosides and -β-D-mannopyranosides. Stereochemical influences of aglycone alcohols. Tetrahedron Lett 1977, 18(2), 175-178.

  1. Page 4, Table 2: The 13C NMR data of 5 should be added to support the β-d-glucosylation- induced shift-trend rule for compounds 1 and 2.

Thank you for your suggestion. We have added it in the table.

  1. Page 5, line 171: Compound 7, and 11→ Compounds 7 and 11

Thank you for your correction. We have revised it.

Reviewer 2 Report

In my opinion, the manuscript needs to be revised before it can be accepted for publication.

Introduction section - the Authors should try to make an effort to emphasize the importance of their studies. They can also dedicate part of Introduction about the activity investigated in the study.

The correct name of studied species is Spilanthes acmella (L.) L.

Results and Discussion is very important part of each manuscript published. In presented manuscript this section is poor and comprises too general explanations. Authors should discuss their results with other scientific papers.

Plant material - Please add the geographical coordinates of occurrence of the tested species. Which reference flora was used to identify the species?

The HPLC conditions are needed.

The conclusions should be integrated with more detailed results summarizing all the study and must reflect the innovation of this study and the perspectives.

Author Response

(The authors gave the same response as above.)

Reviewer 3 Report

The manuscript reports four new compounds (14) along with ten known compounds from the whole plants of Spilanthes acmella and their bone formation activities. Overall, the manuscript is well written and the technologies used in the structural elucidation are satisfactory.

 Followings are minor issues that need to be addressed before acceptance in the Molecules.

Minor comments

  1. Pages 2~8:
  • Numbers of the compounds → bold
  • 2-C-methyl-d-threono-1,4-lactone-3-O-β-d-glucopyranoside (1) →

2-C-methyl-d-threono-1,4-lactone-3-O-β-d-glucopyranoside (1)

  • 2-C-methyl- d-threono-1,4-lactone-3-O- β-d-fructofuranoside (2) →

2-C-methyl- d-threono-1,4-lactone-3-O-β-d-fructofuranoside (2)

  1. Page 3: Table 1
  • compound 1: 1.41 (3H, s) → 1.41 s (3H)
  • compound 2: 1.35 (3H, s) → 1.35 s (3H)
  • compound 3: 2.16 (m), 2.48 (m), 4.27 (2H, m), 1.75 (m), 1.79 (m) → 16 m, 2.48 m, 4.27 m (2H), 1.75 m, 1.79 m
  1. Page 3, line 103: The original reference should be cited instead of ref [25].
  2. Page 4, Table 2: The 13C NMR data of 5 should be added to support the β-d-glucosylation-induced shift-trend rule for compounds 1 and 2.
  3. Page 5, line 171: Compound 7, and 11 → Compounds 7 and 11

Author Response

(The authors gave the same response as above.)
